# Genetic Variability in Carotenoid Contents in a Panel of Genebank Accessions of Temperate Maize from Southeast Europe

**DOI:** 10.3390/plants12193453

**Published:** 2023-09-30

**Authors:** Domagoj Šimić, Vlatko Galić, Antun Jambrović, Tatjana Ledenčan, Kristina Kljak, Ivica Buhiniček, Hrvoje Šarčević

**Affiliations:** 1Agricultural Institute Osijek, Južno Predgrađe 17, 31000 Osijek, Croatia; vlatko.galic@poljinos.hr (V.G.); antun.jambrovic@poljinos.hr (A.J.); tatjana.ledencan@poljinos.hr (T.L.); 2Centre of Excellence for Biodiversity and Molecular Plant Breeding (CroP-BioDiv), Svetošimunska Cesta 25, 10000 Zagreb, Croatia; hsarcevic@agr.hr; 3Faculty of Agriculture, University of Zagreb, Svetošimunska Cesta 25, 10000 Zagreb, Croatia; kkljak@agr.hr; 4Bc Institute for Breeding and Production of Field Crops, Rugvica, Dugoselska 7, 10370 Dugo Selo, Croatia; ibuhinicek@bc-institut.hr

**Keywords:** breeding, carotenoids, genebank, maize, Southeast Europe

## Abstract

Carotenoids are an abundant group of lipid-soluble antioxidants in maize kernels. Maize is a key target crop for carotenoid biofortification focused on using conventional plant breeding in native germplasm of temperate areas traced back partially to traditional cultivars (landraces). In this study, the objectives were to determine the variability of lutein (LUT), zeaxanthin (ZEA), α-cryptoxanthin (αCX), β-cryptoxanthin (βCX), α-carotene (αC), and β-carotene (βC) contents in the grain of 88 accessions of temperate maize from the Croatian genebank, and to evaluate the relationships among the contents of different carotenoids as well as the relationships between kernel color and hardness and carotenoid content. Highly significant variability among the 88 accessions was detected for all carotenoids. On average, the most abundant carotenoid was LUT with 13.2 μg g^−1^ followed by ZEA with 6.8 μg g^−1^ dry matter. A Principal Component Analysis revealed a clear distinction between α- (LUT, αCX, and αC) and β-branch (ZEA; βCX, and βC) carotenoids. β-branch carotenoids were positively correlated with kernel color, and weakly positively associated with kernel hardness. Our results suggest that some genebank accessions with a certain percentage of native germplasm may be a good source of carotenoid biofortification in Southeast Europe. However, due to the lack of association between LUT and ZEA, the breeding process could be cumbersome.

## 1. Introduction

Maize grain contains various types of carotenoids, which are natural pigments responsible for the yellow, orange, and red colors found in many fruits and vegetables. Carotenoids play important roles in plant physiology [1,2], as well as human health [3,4]. The predominant carotenoids found in maize kernels are lutein, zeaxanthin, β-carotene, and cryptoxanthin. Lutein is a yellow pigment and an important dietary carotenoid. It acts as an antioxidant and is known for its role in maintaining eye health, particularly in preventing age-related macular degeneration [5]. Zeaxanthin is another yellow pigment found in maize grain. It is closely related to lutein and has similar benefits for eye health [5]. Zeaxanthin also acts as an antioxidant and is concentrated in the macula of the human retina [6]. β-Carotene is an orange pigment that can be converted into vitamin A in the body. Vitamin A is essential for vision, immune function, and cell growth [7]. Maize grain contains primarily provitamin A carotenoids, which means they can be converted into vitamin A. Cryptoxanthin is an orange-red carotenoid with provitamin A activity and it contributes to the overall carotenoid content of the grain [8]. The specific amounts and ratios of these carotenoids can vary depending on the maize variety, environmental conditions, and maturity stage of the grain [9]. Additionally, carotenoid content can be influenced by processing and cooking methods [10].

Breeding efforts to increase carotenoid content (biofortification) are widely viewed as a valued strategy for sustainably improving the nutritional status of maize [4,11,12,13]. This is particularly important in some areas in Africa where maize is a staple food. Moreover, carotenoid biofortification in maize is also of interest in temperate areas, principally through classical breeding for macular carotenoids (lutein and zeaxanthin) both in adapted [14] and exotic [15] maize germplasm apart from genetic engineering approaches [16]. In Europe, the contents of lutein, zeaxanthin, and total carotenoids were determined in 93 maize landraces of the European Maize Landraces Core Collection (EUMLCC) [17] and showed a higher content of total carotenoids compared with varieties from other countries. The contents of lutein, zeaxanthin, and cryptoxanthin were also characterized in elite Italian and public inbred lines [18]. Revilla et al. [19] pointed out that the Italian inbred varieties selected from traditional maize populations are an appropriate material for improving carotenoid contents in current maize hybrids for the market of traditional maize-based foods. Recently, Calugar et al. [20] analyzed maternal cytoplasm to determine its influence on the carotenoid contents of some inbred lines and hybrids in Romania, while Niaz et al. [21] gave a review on the genetic and molecular basis of carotenoid metabolism in cereals including maize. In Croatia, Zurak et al. [22] analyzed carotenoid contents in commercial maize hybrids giving no breeding recommendations.

The objectives of this study were: (1) to determine the variability of lutein (LUT), zeaxanthin (ZEA), α-cryptoxanthin (αCX), β-cryptoxanthin (βCX), α-carotene (αC), and β-carotene (βC) contents in the grain of 88 accessions of temperate maize from the Croatian genebank, and (2) to evaluate the relationships among contents of different carotenoids as well as the relationships between kernel color and hardness and carotenoid content.

## 2. Results

### 2.1. Introduction

The list of 88 maize accessions (inbred lines) from the Croatian genebank shown in Appendix A also includes the kernel color and kernel type of the accessions. There were 7 accessions with pale yellow, 31 with yellow, 46 with orange, and 4 with deep orange kernels. There were 3 dent, 33 semi-dent, 30 semi-flint, and 22 flint inbred lines.

Using high-pressure liquid chromatography (HPLC), obvious single peaks in chromatograms were detected for LUT ZEA, αCX, βCX, αC, and βC. An example of an HPLC chromatogram is shown in Figure 1.

### 2.2. Analysis of Variance

The analysis of variance (ANOVA) showed that the effect of genotype was highly significant (*p* > 0.01) for all carotenoid compounds (Table 1). The effect of location was significant only for βC at the 0.05 probability level, while the location × genotype interaction was significant for LUT, αCX, βCX, and αC. Repeatability estimates were mostly high from 0.77 for TOT to 0.94 for ZEA.

### 2.3. Mean and Variation in Carotenoid Contents

Mean values for all carotenoid compounds were similar in both locations (Figure 2). Their respective standard deviations indicate high variability among the 88 accessions for all compounds. On average, the most abundant carotenoid was LUT with 13.1 and 13.2 μg g^−1^ followed by ZEA with 6.7 and 6.8 μg g^−1^ in two locations, respectively. Total carotenoids were 24.8 and 25.2 μg g^−1^ in two respective locations. According to respective standard deviations, variations were similar in both locations for all carotenoid compounds, except for the LUT/ZEA ratio (L/Z), where genotypes varied more in Osijek.

There was considerable and significant variability for all carotenoids across the 88 maize accessions averaged over the two locations (Figure 3). For example, LUT ranged from 0.8 to 27.3 μg g^−1^, ZEA from 1.6 to 21.6 μg g^−1^, and TOT from 8.4 to 50.4 μg g^−1^ in dry matter. Altogether, accessions 17, 38, 82, and 85 had the highest total carotenoid contents with more than 40 μg g^−1^, whereby their proportions of LUT and ZEA varied substantially. The proportion LUT/ZEA (L/Z) was 0.44 in accession 17, but 7.66 in accession 38, whereas accessions 82 and 85 had similar L/Z ratios of 1.26 and 0.91, respectively. On the other hand, accession 23 had the lowest TOT contents with only 0.9 and 1.7 μg g^−1^ of LUT and ZEA, respectively.

### 2.4. Multivariate Analyses

A Principal Component Analysis (PCA) on the data of carotenoid contents was performed to summarize multivariate similarities among the maize accessions. The principal components (PC1 and PC2) accounted for 69.8% of the total variation (Figure 4). The position of maize inbred lines along the PC1 in the PCA biplot was mainly defined by ZEA, βXC, and βC. There was a clear distinction between α- (LUT, αCX, and αC) and β-branch (ZEA; βXC and βC) carotenoids. β-branch carotenoids had negative loadings indicating a negative correlation with the PC1. The position of the accessions along PC1 was set according to β-branch carotenoids including the accessions with the highest TOT contents (17, 82, and 85) placed at the far left. The position of the maize inbred lines along PC2 was set primarily according to LUT and αCX, whereby the accession with the lowest TOT content was positioned opposite to the LUT and αCX vectors at the bottom of the biplot.

Figure 5 presents the results of a k-means clustering analysis performed on the carotenoid contents in maize kernels from 88 accessions of temperate maize. The data points in the figure represent individual maize accessions, and they are colored according to the clusters accordingly. The larger points in the figure represent the centroids of each cluster, which are the average values of the data points within each cluster. The clustering analysis revealed two distinct groups among the maize accessions based on their carotenoid contents. This suggests that there are significant differences in carotenoid contents among these accessions.

### 2.5. Correlations of Carotenoid Contents with Kernel Color and Kernel Hardness 

The distribution of the contents of six carotenoids in the kernels of maize inbred lines grouped according to the color of the kernel (PY = pale yellow, Y = yellow, O = orange, and DO = deep orange) together with the Pearson correlation coefficients between the color intensity of the kernel and the carotenoid contents is shown in Figure 6. A significant, weak positive correlation was observed between kernel color and the contents of all carotenoids except βC, with correlation coefficients ranging from 0.21 (αC) to 0.27 (αCX). The mean contents of all carotenoids except βC were lowest for the pale yellow class and highest for the deep orange class, but the relative differences between the two color classes were more pronounced for the α-branch carotenoids (LUT, αCX, and αC), showing a more than twofold increase (Figure 6). However, the large variation within color classes for all carotenoids (Figure 6) resulted in relatively weak observed correlations between kernel color and carotenoid contents. Compared with the individual carotenoids, the correlation between total carotenoid contents and kernel color was stronger (r = 0.43), and a twofold increase was observed from the pale yellow to the deep orange class, i.e., from 16.34 to 32.81.

The distribution of the contents of six carotenoids in the kernels of maize inbred lines grouped according to the kernel type (D = dent, S-D = semi-dent, S-F = semi-flint, and F = flint) together with the Pearson correlation coefficients between the kernel hardness and the carotenoid contents is shown in Figure 7. A significant, weak positive correlation was observed between kernel hardness and β-branch carotenoid contents, with correlation coefficients of 0.31, 0.23, and 0.25 for ZEA, βCX, and βC, respectively. The mean content of the three carotenoids was lowest in the dent class and highest in the flint class, with the most pronounced relative increase between the two kernel types observed for ZEA (80%) (Figure 7). However, as with the color classes, the large variation within the kernel types for all carotenoids (Figure 7) resulted in relatively weak observed correlations between kernel hardness and carotenoid contents. Although the correlation between kernel hardness and αCX contents was not significant, it tended to be negative, with the lowest mean content observed for flints. The correlation coefficient between total carotenoid contents and kernel hardness (r = 0.27) was of the same order of magnitude as observed correlation coefficients for individual β-branch carotenoids.

## 3. Discussion

Although there was interest in carotenoids almost for a century in maize [23], only using high-pressure liquid chromatography (HPLC) was it feasible to conduct effective phenotypic selection for higher levels of carotenoids [14]. To date, several studies have shown substantial genotypic variation in maize for the most important carotenoids such as LUT, ZEA, βCX, and βC. The ranges were different depending on the diversity and background of the analyzed germplasm. Our results are comparable to those presented in [14,18], where temperate maize material was evaluated using HPLC. The variation in LUT contents in our study was greater than those in the two referenced studies. On the other hand, ZEA, βCX, and βC contents were similar. On average, there was a twofold larger content of lutein than zeaxanthin with a high LUT/ZEA ratio of >2. In Italian inbred lines, the ratio was 0.6, where average lutein content was only 7.1, but zeaxanthin content was 11.7 [18]. Generally, the contents of macular carotenoids can be much higher after selection reaching 101 μg g^−1^ for LUT and 65 μg g^−1^ for ZEA [15]. However, in our study, significant location × genotype interaction was detected for LUT, αCX, βCX, and αC. In a more comprehensive investigation, Muzhingi et al. [12] showed that the interaction is mostly not significant. Repeatability estimates, which are equivalent to the heritability estimates on an entry-mean basis [24], were comparable to the heritabilities presented by Diepenbrock et al. [25] that ranged from 0.7 for αC to 0.9 for LUT, ZEA, and βCX.

The high variability in carotenoids in different maize accessions was also noticeable via PCA (Figure 4). Strong associations among beta carotenoids (ZEA, βCX, and βC) were presented on the biplot of PCA with a clear distinction from alpha carotenoids (LUT, αCX, and αC). Similar patterns among the accessions were observed by analyzing k-means clustering. The results from multivariate analyses presented in Figure 4 and Figure 5 can provide valuable insights for efforts aimed at the biofortification of maize with carotenoids by stratifying maize accessions that have similar carotenoid profiles. Roughly, the maize genotypes were grouped in two clusters made via k-means analysis according to their respective contents of alpha and beta carotenoids. These groups can be further investigated to identify promising candidates for breeding programs.

The tightest positive correlation was between ZEA and βCX (r = 0.75), followed by the correlation between βCX and βC (r = 0.48). There was no correlation between macular carotenoids LUT and ZEA (r = −0.16). In contrast, Muthusamy et al. [26] presented a moderate (r = 0.53) to strongly positive (r = 0.97) correlation between LUT and ZEA in two panels of traditional Indian and biofortified inbred lines, respectively. Remarkably, they also found no correlation between βCX and βC. This study demonstrated that βC did not correlate with kernel color. Contrarily, kernel color was positively correlated with LUT, ZEA, βCX, and TOT in a range from r = 0.25 (LUT) to r = 0.47 (ZEA). The results of the present study are in agreement with Muthusamy et al. [26], in which the contents of all carotenoids except βC showed significant positive, although weak correlations with kernel color, with correlation coefficients ranging from 0.21 (αC) to 0.27 (αCX). On the other hand, in the U.S. maize nested association mapping panel [27], βC as well as ZEA and βCX were correlated with kernel color (r = 0.53, 0.76, and 0.66, respectively), while the corresponding correlations for LUT and αC were negligible. The correlation between kernel color and total carotenoids (r = 0.43) was in the present study stronger than for individual carotenoids and was similar to the corresponding correlation (r = 0.47) found by Muthusamy et al. [24], but was lower than that reported by La Porte et al. [27] (r = 0.69). The previously reported positive phenotypic correlation between carotenoid contents and kernel color has also been supported by genetic studies, which reported several genes with significant pleiotropy between kernel color and one or more carotenoid traits [27,28].

In addition to kernel color, kernel hardness has also been associated with carotenoid contents in maize kernels [29,30,31]. Although kernel hardness has generally been associated with an improved ability to store more total carotenoids [30,31], carotenoid profile is also influenced by kernel type [29,31], with flint genotypes having higher levels of β-branch carotenoids (ZEA, βCX, and βC), while dent genotypes have higher levels of α-branch carotenoids (LUT, αCX, and αC). This is also supported to some extent by the results of the present study, in which a weak but significant positive correlation was observed between kernel hardness and the contents of ZEA, βCX, βC, and total carotenoids (r = 0.31, 0.23, 0.25, and 0.27, respectively).

Maize accessions in our study were predominantly inbred lines traced back to deteriorated populations developed by breeders during the first half of the 20th century, and some of them traced back to traditional cultivars (landraces) of uncertain origin, all referring to native germplasm. The percentage of native germplasm ranged from 0 to 100% and no particular native germplasm had generally high carotenoid contents. The accessions 20, 29, 67, and 80 having 100% of native germplasm had just average carotenoid contents. However, accessions 17 and 38 having 47% of native germplasm were among the four accessions with the highest TOT values, but with considerably different L/Z ratios. Nevertheless, these two inbred lines, tracing back from the local inbred lines L131F-e and L131F-d, can be considered a good starting point for breeding macular carotenoids which eventually could have nutritional benefits to consumers. Specifically, line 38 had the highest LUT content in our experiment reaching 27.31 μg g^−1^.

Our results suggest that several maize genebank accessions with a certain percentage of native germplasm may be a good source of carotenoids for biofortification breeding programs in Southeast Europe. This is particularly true for the accessions of semi-flint or flint types of kernel. However, due to the lack of association between macular carotenoids of LUT and ZEA in accessions with high total carotenoid contents, the breeding process could be cumbersome. The most important issues for applied breeding programs would be the choice of the right integration method into elite maize breeding material [32] as well as a choice of suitable heterotic patterns, which are different compared with Western Europe [33]. Methods of molecular breeding including analysis of genetic diversity and selection signatures [33] could be helpful during the breeding process.

## 4. Materials and Methods

Eighty-eight temperate maize inbred lines included in the present study are part of the collection of maize accessions maintained at the University of Zagreb, Faculty of Agriculture, within the Croatian genebank [34].

The inbreds have different proportions of native germplasm (maize landraces and inbred lines from Southeastern Europe) in their pedigrees (Appendix A) including varieties traced back from the first half of the 20th century (“Šidski zuban”, “Novosadski Fleischman”) [35], as well as obsolete inbred lines developed in the region (L131F lines, L1-26, L86, etc.). Otherwise, the origin of the material is either from the U.S. Corn Belt including open-pollinated varieties such as Lady Finger, synthetics (Illinois Syn60c, Pioneer synthetics), and inbred lines (e.g., B87, H99, Mo17, and Pa492), or populations and inbred lines from Western Europe (Lacaune, F2) and Eastern Europe (Russian synthetics).

A field trial with 88 maize inbred lines set up as a randomized complete block design in two replicates was conducted at locations Osijek and Zagreb in Croatia in 2019. The experimental plots consisted of a 4 m row with 0.70 m of spacing between rows and were machine-planted at a density of approximately 50,000 plants per hectare. Six to eight plants in each plot were self-pollinated to avoid xenia effects. After maturity, grains from individual ears of self-pollinated plants were harvested and bulked at shelling. Inbreds were visually scored for their kernel color and kernel type on a bulk sample of 100 kernels per plot using the IBPGR maize descriptor list [36]. Grain samples were stored in the dark at 4 °C until carotenoid extraction to avoid loss of carotenoids.

Carotenoids from whole maize grain were extracted and quantified according to the procedure described by Kurilich and Juvik [37], using β-apo-carotenal as an internal standard. For the analysis, the maize samples were ground to pass through a 0.3 mm sieve (Cyclotec 1093, Foss Tocator, Hoganas, Sweden). Then, the samples (0.6 g) were ultrasonicated (10 min; Sonorex TK 52, Bandelin, Berlin, Germany) and homogenized (1 min per sample; T10 Ultra-Turaxx, IKA, Staufen, Germany) with 6 mL of ethanol containing 0.1% of butylhydroxytoluene (BHT). The samples were then placed in a water bath and incubated for 5 min at 85 °C. Subsequently, the samples were saponified with 100 μL of 80% KOH for 10 min at 85 °C. After the samples were cooled by adding 3 mL of chilled ultrapure water and placing them in an ice bath, carotenoids and tocols were extracted with hexane in aliquots of 3 mL. The phases were separated via centrifugation for 10 min at 2200× *g* (Centric 322A, Tehtnica, Slovenia). The extraction procedure was repeated until a colorless upper hexane layer was achieved. The collected supernatants were evaporated using a rotary vacuum concentrator (RVC 2-25CD plus, Martin Christ, Germany) and dissolved in 200 µL of acetonitrile:dichloromethane:methanol (45:20:35, *v*/*v*/*v*) containing 0.1% BHT. Extractions were carried out under dim light, and extracts were analyzed further using HPLC on the same day.

Carotenoids in prepared extracts were separated and quantified using a SpectraSystem HPLC instrument (Thermo Separation Products, Inc., Waltham, MA, USA) equipped with a quaternary gradient pump (P4000), an autosampler (AS3000),) and a UV-Vis detector (UV2000). Compounds were separated on two sequentially connected C18 reversed-phase columns: a Vydac 201TP54 column (5 μm, 4.6 × 150 mm; Hichrom, Reading, UK) followed by a Zorbax RX-C18 column (5 μm, 4.6 × 150 mm; Agilent Technologies, Santa Clara, CA, USA). The separation columns were protected by a Supelguard Discovery C18 guard column (5 μm, 4 × 20 mm; Supelco, Bellefonte, PA, USA). The mobile phase consisted of acetonitrile:methanol:dichloromethane (75:25:5, *v*/*v*/*v*) containing 0.1% BHT and 0.05% triethylamine. An aliquot of 30 μL was injected, and the flow rate was 1.8 mL/min. The separations were performed at room temperature. Carotenoids were monitored on a UV-Vis detector at 450 nm.

Separated compounds were identified by comparing their retention times and quantified using external standardization with calibration curves using commercially available standards (r^2^ ≥ 0.99). Carotenoid standards (LUT, ZEA, αCX, βCX, αC, and βC; purity ≥ 98%) were obtained from Extrasynthese (Genay, France). Carotenoid content was expressed as mean ± standard deviation of two field replications. The total carotenoid content (TOT) was calculated by summing the contents of the individual carotenoids.

ANOVA was conducted using the PLABSTAT software (version 1997) [38]. Repeatability (equivalent to the heritability on an entry mean basis [22]) was estimated as a measure of the precision of trials expected from multiple measurements as follows:r = V_G_/[V_G_ + V_L×G_/l + V_error_/(lr)]
where V_G_, V_L×G_, and V_error_ are components of the genotypic variance, the location × genotype interaction variance, and error variance, respectively, whereas l and r are the number of locations and replicates, respectively.

Principal Component Analysis (PCA) was used for screening variability by finding the synthetic variables, i.e., principal components (PCs) calculated as linear combinations of original variables. Individual PCs represent linear statistical models with the scores (distance from the PC origin for every data point), loadings (variable contributions for each PC), and residuals. All input variables were scaled, centered, and log-transformed. Analyses were carried out using the R/prcomp function and the biplots were created with custom ggplot2 scripts [39]. Likewise, k-means clustering was carried out using R/cluster library [39]. A significant number of clusters was determined using the “elbow” heuristic method (not shown) which converged at 2 clusters. Clusters were plotted using a R/ggplot2 library [39].

## 5. Conclusions

Our results suggest that genetic variability of carotenoid contents in our panel of genebank accessions of temperate maize from Southeast Europe was stratified mainly according to the two groups of α- (LUT, αCX, and αC) and β-branch (ZEA; βCX and βC) carotenoids whereby the origin of accessions played a minor role. Further, associations between kernel color and hardness with carotenoid contents were weaker than expected.

## Figures and Tables

**Figure 1 plants-12-03453-f001:**
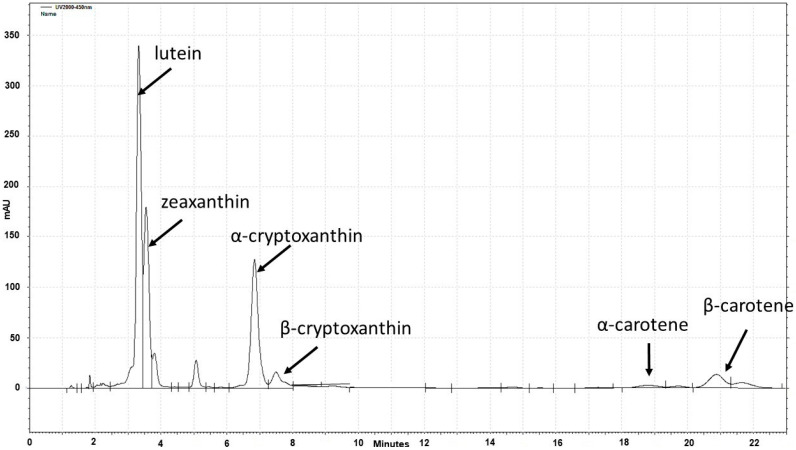
An example of an HPLC chromatogram of carotenoids extracted from one of the analyzed maize samples.

**Figure 2 plants-12-03453-f002:**
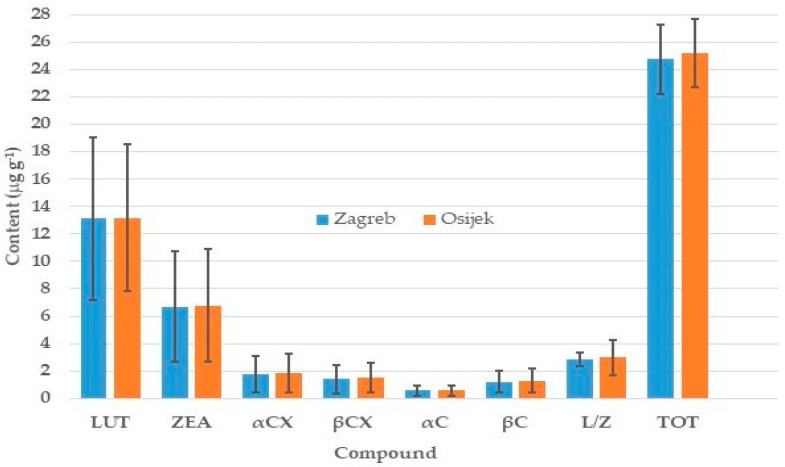
Mean values for contents of carotenoid compounds in 88 maize inbred lines grown at locations Zagreb and Osijek in 2019. Vertical bars denote respective standard deviations. The acronyms LUT, ZEA, αCX, βCX, αC, βC, LUT/ZEA, and TOT indicate lutein, zeaxanthin, α-cryptoxanthin, β-cryptoxanthin, α-carotene, β-carotene, lutein/zeaxanthin ratio, and total carotenoid contents, respectively.

**Figure 3 plants-12-03453-f003:**
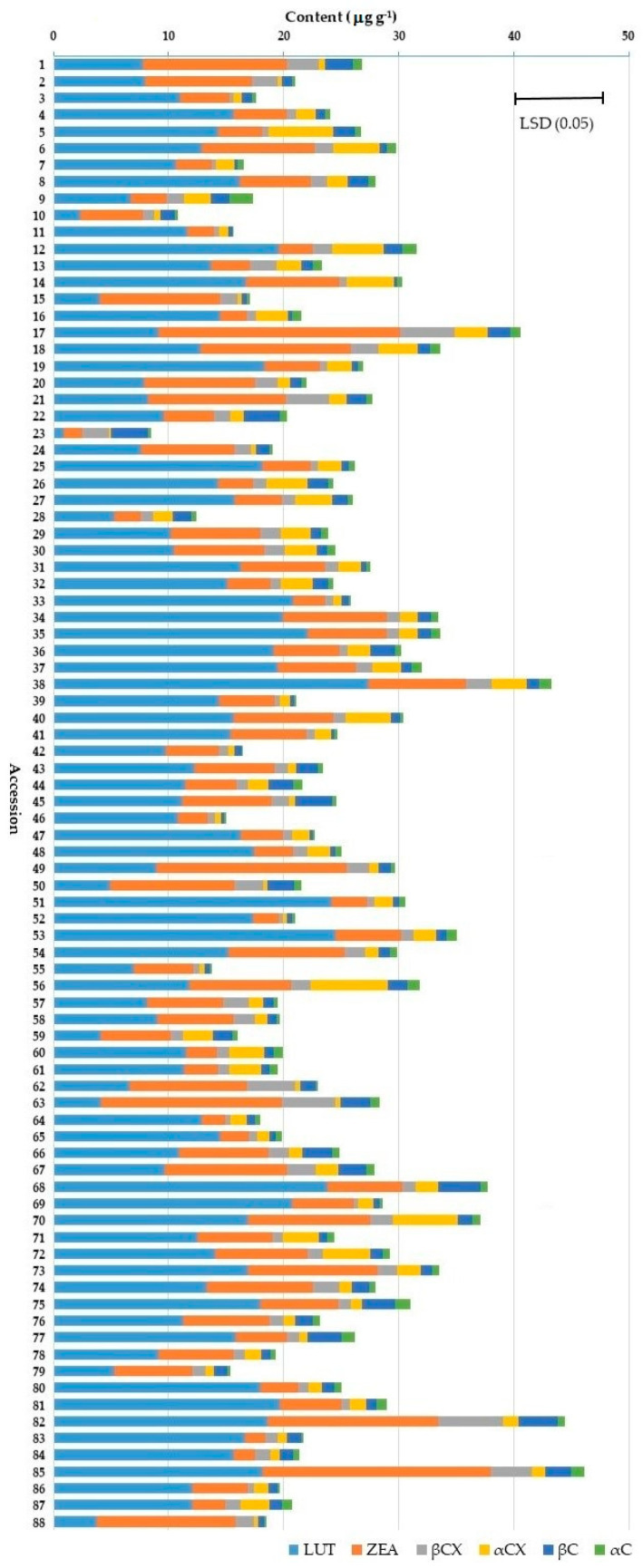
Carotenoid profiles of 88 maize genebank accessions averaged over two locations. The error bar (upper right) shows the common least significant difference at the 0.05 probability level (LSD (0.05)) on total carotenoid contents. The acronyms LUT, ZEA, αCX, βCX, αC, βC, LUT/ZEA, and TOT indicate lutein, zeaxanthin, α-cryptoxanthin, β-cryptoxanthin, α-carotene, β-carotene, lutein/zeaxanthin ratio, and total carotenoid contents, respectively.

**Figure 4 plants-12-03453-f004:**
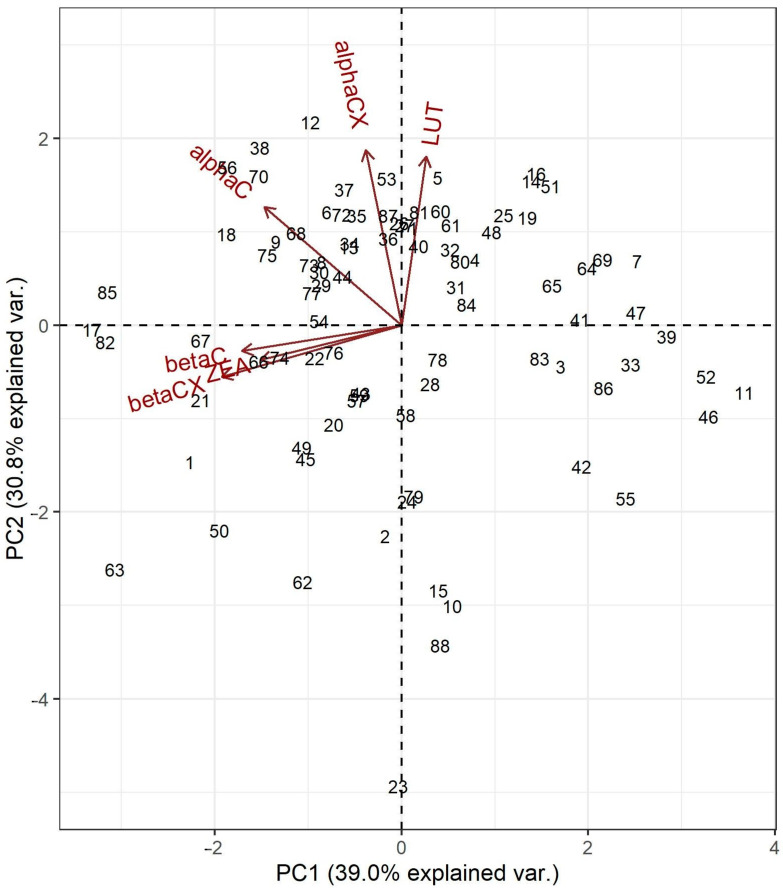
Biplot of Principal Component Analysis (PCA) based on six carotenoids measured in 88 maize genebank accessions.

**Figure 5 plants-12-03453-f005:**
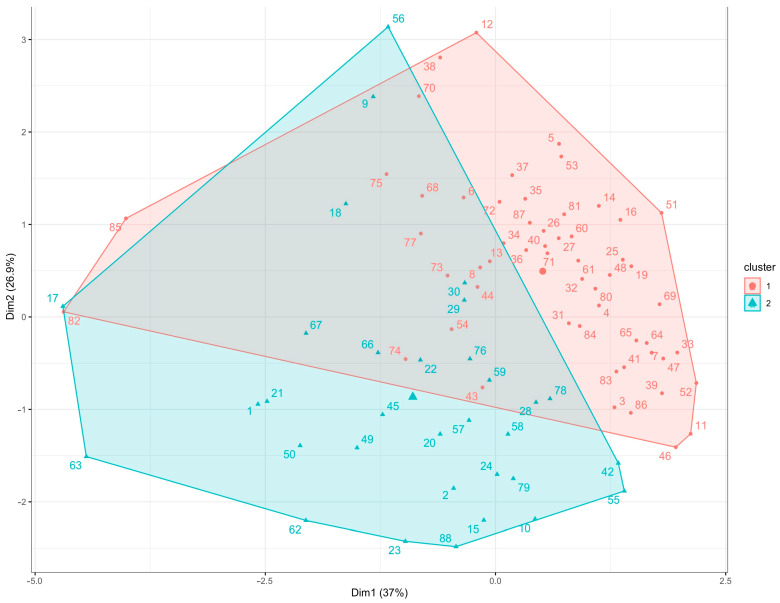
K-means clustering of carotenoid contents in maize kernels from 88 accessions of temperate maize. The two clusters are represented by different colors, with the larger dots and triangles indicating the centroids of each cluster.

**Figure 6 plants-12-03453-f006:**
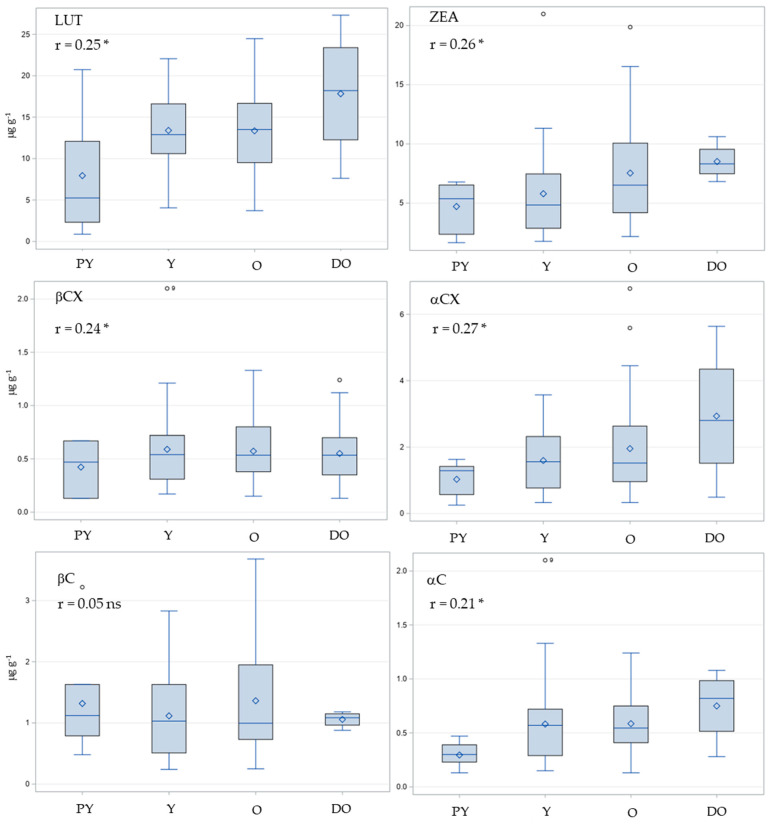
Distribution of the contents of six carotenoids (LUT = Lutein, ZEA = Zeaxanthin, βCX = β-cryptoxanthin, αCX = α-cryptoxanthin, βC = β-carotene, and αC = α-carotene) in the kernels of maize inbreds grouped by kernel color (PY = pale yellow, Y = yellow, O = orange, and DO = deep orange). * Pearson correlation coefficient (r) significant at the 0.05 probability level; ns r not significant.

**Figure 7 plants-12-03453-f007:**
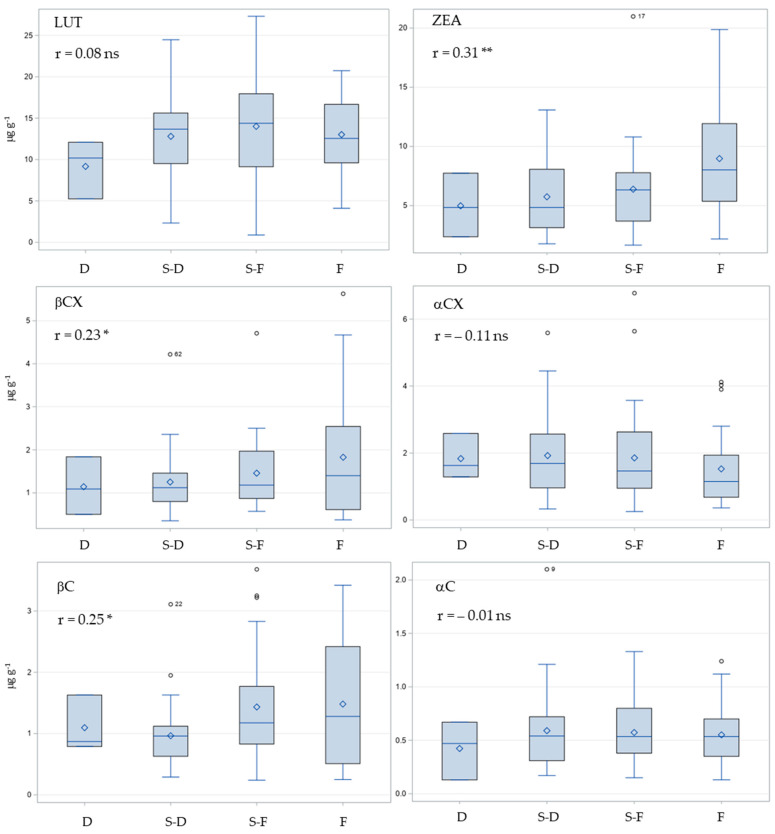
Distribution of the contents of six carotenoids (LUT = Lutein, ZEA = Zeaxanthin, βCX = β-cryptoxanthin, αCX = α-cryptoxanthin, βC = β-carotene, and αC = α-carotene) in the kernels of maize inbreds grouped by kernel hardness (D = dent, S-D = semi-dent, S-F = semi-flint, and F = flint). * and ** Pearson correlation coefficient (r) significant at the 0.05 and 0.01 probability levels, respectively; ns r not significant.

**Table 1 plants-12-03453-t001:** Mean squares of the analysis of variance for carotenoid compounds in 88 maize inbred lines from genebank.

		Mean Squares							
Source	df	LUT	ZEA	αCX	βCX	αC	βC	LUT/ZEA	TOT
Location(L)	1	0.1	0.4	0.2	0.4	0.0	0.4 *	0.6	7.2
Genotype(G)	87	58.3 **	31.6 **	3.4 **	2.0 **	0.2 **	1.4 **	9.2 **	103.9 **
L × G	87	6.0 **	1.4	0.4 +	0.2 *	0.03 **	0.1	0.7	10.9
Error	198	3.5	1.9	0.3	0.1	0.02	0.1	1.7	27.9
Repeatability		0.90	0.94	0.88	0.90	0.90	0.93	0.83	0.77

+, *, ** F test of corresponding mean squares significant at the 0.1, 0.05, and 0.01 probability levels, respectively. The acronyms LUT, ZEA, αCX, βCX, αC, βC, LUT/ZEA, and TOT indicate lutein, zeaxanthin, α-cryptoxanthin, β-cryptoxanthin, α-carotene, β-carotene, lutein/zeaxanthin ratio, and total carotenoid contents, respectively.

## Data Availability

The data were obtained from the Faculty of Agriculture, University of Zagreb, and the Agricultural Institute Osijek. Data are available on request from the corresponding author with the permission of the Faculty of Agriculture, University of Zagreb, and the Agricultural Institute Osijek.

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
