# Peer review of "Genetic Variability in Carotenoid Contents in a Panel of Genebank Accessions of Temperate Maize from Southeast Europe"

_plants, 2023, doi:10.3390/plants12193453_

Round 1

Reviewer 1 Report

This report has relied on data from an outside source see fig 2 and supplementary information. Can I assume permissions have been received and the 3rd party have approved publication..

Also the current methodology is a statistical examination of this outside data.

There is needed inclusion in the report on how data for figs 1 and 2 has been obtained.

Also regarding  figs 4 and 5 information on how colour and hardness were determined

Also Fig 4 has no units for the x axis (this may have been corrected by now!)

My ethical concern is that the original data, that could very well be accurate, was secondary to the works undertaken by the authors and had permission been received to publish this information or has it been published elsewhere..

Regarding the introduction this should reflect new inclusions

Author Response

  1. This report has relied on data from an outside source see fig 2 and supplementary information. Can I assume permissions have been received and the 3rd party have approved publication.

RESPONSE: The data presented in this study are not external data obtained from third parties. These data are from the field trials and subsequent laboratory analyses conducted by authors of the manuscript. Some authors are members of the „Cereals and Maize Working Group within the Croatian Gene bank and the material (88 inbred lines) was provided by Hrvoje Šarčević (co-author), who has chaired the working group since 2007 and is the head of the maize collection at the Faculty of Agriculture. Three co-authors of the manuscript (Šimić, Buhiniček and Šarčević) are the Croatian representatives in the Maize Working Group within ECPGR (see .https://www.ecpgr.cgiar.org/contacts-in-ecpgr/ecpgr-contacts/maize )

  1. Also the current methodology is a statistical examination of this outside data.

RESPONSE: See 1.

3.  There is needed inclusion in the report on how data for figs 1 and 2 has been obtained.

RESPONSE:  See 1.

  1. Also regarding  figs 4 and 5 information on how colour and hardness were determined

RESPONSE: Inbreds were visually scored for their kernel colour and hardeness using the IBPGR maize descriptor list. The sentence has been added to the Material and methods chapter to explain this (lines 294-296).

  1. Also Fig 4 has no units for the x axis (this may have been corrected by now!)

RESPONSE: We added units in Fig. 4.

  1. My ethical concern is that the original data, that could very well be accurate, was secondary to the works undertaken by the authors and had permission been received to publish this information or has it been published elsewhere.

RESPONSE: We presented the original data from the experiment we conducted ourselves. See 1.

  1. Regarding the introduction this should reflect new inclusions

RESPONSE:  Two recent and important references are added (lines 60-63).

Reviewer 2 Report

Genetic Variability of Carotenoid Contents in a Panel of Genebank Accessions of Temperate Maize from Southeast Europe is overall well written article. Authors analyzed the carotenoid and xanthophyll variations among different maize varities.

I have minor suggestions and questions to improve the article.

1) Is the carotenoid content variation affected by the extraction method? because the xanthophylls and carotenoids have polarity difference. I am wondering how authors normalize this polarity effect.

2) It would be great to include a  HPLC chromatogram with  carotenoid separation.

3) line 149 and line 171 typo A-cryptoxanthin should be corrected to the alpha cryptoxanthin.

Author Response

1) Is the carotenoid content variation affected by the extraction method? because the xanthophylls and carotenoids have polarity difference. I am wondering how authors normalize this polarity effect.

RESPONSE: The method used for extraction was chosen since allows simultaneous extraction of both carotenoids and tocols from maize grain. The only modification was that extraction with hexane was repeated until colourless extract (usually six extractions compared to three extractions in method by Kurilich and Juvik) to ensure complete carotenoid extraction. In addition, samples were ground to pass 0.3 mm screen, and homogenised with ethanol before the hot saponification, and this homogenisation included ultrasonication for 10 minutes and homogenisation with T10 Ultra-Turaxx. These additional steps were added while implementing the method; method was compared with extraction with several different solvents and their mixtures (acetone, acetone-ethanol-hexane, rehydration of samples before extraction) and results were similar. In accordance with these additional steps, sentences in lines 299-305 are amended

2) It would be great to include a HPLC chromatogram with  carotenoid separation.

RESPONSE: An example of one chromatogram is added (Figure 1). The new text is in lines 77-79.

3) line 149 and line 171 typo A-cryptoxanthin should be corrected to the alpha cryptoxanthin.

RESPONSE: The typos are corrected.

Reviewer 3 Report

The presente document descrives the carotenoids content and composition of a panel of corn accession.

The introduction is complete and provides a good inshight on the topic, also presents clearly the problematic.

The material and method section is also clear, the only thing that could be implemented a little more is about the origin of the material (a little is said in the discussion, but could be completed in the M & M section too.

The results are globally well presented but maybe it will be interesting to add a clustering based both in the content and or the composition of carotenoids of the genotypes in order to proceed to any kind of breeding, and link it to the origin of the material.

The discussion is complete.

You need to include a conclusion to your study.

Minor corrections can be found in the attached document

Author Response

The presente document descrives the carotenoids content and composition of a panel of corn accession.The introduction is complete and provides a good inshight on the topic, also presents clearly the problematic.The material and method section is also clear, the only thing that could be implemented a little more is about the origin of the material (a little is said in the discussion, but could be completed in the M & M section too.

RESPONSE: The origin of the material is added, lines 281-288.

The results are globally well presented but maybe it will be interesting to add a clustering based both in the content and or the composition of carotenoids of the genotypes in order to proceed to any kind of breeding, and link it to the origin of the material.

RESPONSE: K-means clustering is added (Figure 5) and commented (lines 133-144 and 223-229)

The discussion is complete.

You need to include a conclusion to your study.

RESPONSE: Conclusion is added.

Minor corrections can be found in the attached document

  1. Please check the sentence

RESPONSE: The sentence is corrected.

  1. please indicate in the footnote the acronyms used in the table to make it independent for the lecture

RESPONSE: The acronyms are indicated.

  1. we do not see error bar, is this normal Have you been able to classe the varieties or generate groups of common composition?

RESPONSE: Error bar was present and it is additionally described in the figure caption

  1. It will be interesting to perform some clustering of the varieties

RESPONSE: K-means clustering is added (Figure 5) and commented (lines 133-144 and 223-229)

Round 2

Reviewer 1 Report

I am satisfied with the changes. I note that some the pigments a referenced to health benefits.

Overall, do the authors have any suggestions as to which of pigmented maize individuals have greater health and nutritional benefits to consumers?

Author Response

REVIEWER'S COMMENT

I am satisfied with the changes. I note that some the pigments a referenced to health benefits. Overall, do the authors have any suggestions as to which of pigmented maize individuals have greater health and nutritional benefits to consumers?

RESPONSE:  We added two sentences (lines 268-272) in this regard. 

Reviewer 3 Report

The changes requested have been made, only in Figure 2 is needed to add the acronyms mean in the legend; the quality of image is not good

Author Response

REVIEWER'S COMMENT

The changes requested have been made, only in Figure 2 is needed to add the acronyms mean in the legend; the quality of image is not good

RESPONSE: The acronyms are added in Figure 2. The same is done in Figure 3. A new version of the image will be provided if necessary.